# Characterisation of the R2R3 Myb subgroup 9 family of transcription factors in tomato

**Gwendolyn V. Davis**[1,2], **Beverley J. Glover**[2]*

**1** Department of Life Sciences, University of Warwick, Coventry, United Kingdom, **2** Department of Plant Sciences, University of Cambridge, Cambridge, United Kingdom

* bjg26@cam.ac.uk

## Abstract

Tomato (*Solanum lycopersicum)* has many epidermal cell outgrowths including conical petal cells and multiple types of trichomes. These include the anther-specific trichome mesh which holds the anthers connate. The R2R3 Myb Subgroup 9 family of transcription factors is involved in development of epidermal cell outgrowths throughout the angiosperms. No previous study has examined all members of this transcription factor family in a single species. All 7 *R2R3 Myb Subgroup 9* genes were isolated from tomato. They were ectopically expressed in tobacco to assess their ability to induce epidermal cell outgrowth. Endogenous expression patterns were examined by semi-quantitative RT-PCR at different stages of floral development relative to the development of anther trichomes. We report variation in the degree of epidermal cell outgrowth produced in transgenic tobacco by each ectopically expressed gene. Based on expression profile and ectopic activity, *SlMIXTA-2* is likely involved in the production of leaf trichomes. *SlMIXTA-2* is expressed most strongly in the leaves, and not expressed in the floral tissue. *SlMYB17-2* is the best candidate for the regulation of the anther trichome mesh. *SlMYB17-2* is expressed strongly in the floral tissue and produces a clear phenotype of epidermal cell outgrowths when ectopically expressed in tobacco. Analysis of the phenotypes of transgenic plants ectopically expressing all 7 genes has revealed the different extent to which members of the same transcription factor subfamily can induce cellular outgrowth.

## Introduction

The R2R3 Myb proteins are a plant specific family of transcription factors which contain two copies of the Myb DNA binding domain [1] and carry out plant-specific functions [2–4] The family can be divided into subgroups based on other conserved motifs usually external to the Myb domain [4, 5]. Some of these subgroups have been shown to regulate specific functions and phenotypes.

The R2R3 Myb Subgroup 9 subfamily of proteins is an ancient lineage [6] that is especially important in the control of epidermal cell modification. Members of this protein family have the R2R3 Myb DNA binding domain composed of the R2 and R3 repeats [7] and also share their own subgroup 9 domain which forms a conserved motif around 30 peptides downstream

**Data Availability Statement:** All relevant data are within the paper and its Supporting Information files.

**Funding:** GVD was supported by the Natural Environment Research Council (grant number NE/

L002507/1). https://www.ukri.org/councils/nerc/.
The funders had no role in study design, data
collection and analysis, decision to publish, or
preparation of the manuscript.

from the second MYB repeat [5]. Duplication within this subgroup early in seed plant evolution led to the creation of two gene lineages: 9A and 9B [6]. These lineages underwent further duplication and sub functionalisation, leading to four clades of genes within the R2R3 subgroup 9 family in the eudicots [6]. These four subclades were named by Brockington et al. (2013): subgroup 9A: *MIXTA* and *MIXTA-like;* and subgroup 9B: *Myb17* and *Myb17-like*. Subgroup 9A and 9B proteins have been shown to perform a range of functions in the control of epidermal cell outgrowth, but the roles of the subgroup 9B proteins are less well understood than those of subgroup 9A [6].

R2R3 Myb Subgroup 9 proteins are involved in the regulation of expression of genes involved in directional cell outgrowth: producing trichomes, conical cells and papillae. These epidermal cell outgrowths have been shown, in some cases, to follow the same developmental pathway, with differences in the timing of expression resulting in the different morphologies. For example, if cell division has finished when cell outgrowth occurs then expression can result in conical cells, however expression during cell division may result in the development of multicellular trichomes [8]. A role for these genes in the regulation of epidermal cell outgrowth has been demonstrated in a variety of species and shown to be conserved throughout the angiosperms in *Antirrhinum majus* [9], *Petunia hybrida* [10], *Arabidopsis thaliana* [10], *Thalictrum thalictroides* [11], cotton fibre initiation and development ([12] and elongation [13], *Mimulus guttatus* [14], *Medicago truncatula* [15] and in *Lotus japonicus* [6]. R2R3 Myb subgroup 9 genes were first shown to be involved in epidermal cell outgrowth in *Antirrhinum majus* where the expression of the subgroup 9A gene *MIXTA* in the petal epidermis was found to be necessary for the formation of conical cells [9]. It was further shown that the *MIXTA* gene was also sufficient for the production of conical cells when ectopically expressed in *Antirrhinum* and in heterologous hosts [8, 16]. However, despite the wealth of studies examining individual members of this transcription factor family, no study has examined all members of the subgroup 9 family from a single species and so it has not been possible to date to draw conclusions about relative functions and phenotypic effects of the different members of the family.

*Solanum lycopersicum* (tomato) is an economically important crop plant. All species of the genus *Solanum* are buzz pollinated. Tomatoes produce multiple types of trichomes on all epidermal surfaces of the plant and there is considerable diversity in the density, morphology and chemical composition of trichomes [17]. Three main types of glandular trichomes have been described in *Solanum lycopersicum*, (type I, VI and VII) as well as two types of non-glandular trichomes (II and III), [18]. An additional type of glandular trichome, type IV, is abundant in the wild tomato species *S. pennellii*, but is absent in cultivated *S. lycopersicum*, despite the close relationship between these species [18, 19]. Each of the types of trichomes found in tomato are morphologically distinct. Of the glandular trichomes, type I have a multicellular base and long multicellular stalk (approximately 2mm) with a small glandular tip at the end. The type IV trichomes are shorter with a unicellular base. Type VI trichomes also have a shorter multicellular stalk (approximately 0.1mm) but with a four celled glandular tip. Type VII trichomes are even shorter (less than 0.05mm) with a unicellular stalk and a glandular tip consisting of between four and 8 cells [18]. The non-glandular trichomes types found in tomato (II and III) are similar in length, at between 0.2 and 1mm. However they differ in that Type II have a multicellular base while type III have a unicellular base [18]. Trichomes are important in resistance to herbivore attack, by providing mechanical resistance that obstructs the movement of arthropod herbivores [20, 21], making the plant less palatable [22, 23] and, in the case of glandular trichomes, producing secondary metabolites for toxicity or entrapment of herbivores [24–27].

The trichomes of tomato play an additional, highly unusual, role. In tomato the anthers are held together in a connate structure by a mesh of interlocking trichomes [28]. These trichomes are multicellular but short and non-glandular, they are also distinct morphologically from the

other trichome types found elsewhere on tomato. This fused 'pepper pot' anther cone is generally uncommon in the genus *Solanum* but is found in all members of the Tomato subclade of *Solanum* (including all tomato wild-relatives such as *S. pimpinellifolium* and *S. penellii*). *Solanum* flowers are pollinated by pollen-gathering bees, which sonicate the flowers to release pollen from the pores at their tips, a process known as buzz pollination. The fused cone of anthers in tomato results in all the anthers being sonicated together as a single unit during buzz pollination. This is hypothesised to increase pollination efficiency and pollinator foraging efficiency [28].

Conical cells are another form of epidermal cell outgrowth found in tomato. Conical cells play an important role on the petals of many flowers [29–31], where they provide improved grip to pollinating insects interacting with the flower [31, 32]. Conical cells can be found on the petals of tomato, although they have been lost in some other species of the genus *Solanum* [33]. The anther trichomes of tomato are unicellular and strongly resemble elongated conical cells, further supporting a developmental link between these specialised epidermal cell types.

Previous work has identified functions for some members of the Myb Subgroup 9 family of transcription factors in tomato prior to this study. The most studied of these genes is named in this study *SlMIXTA-like-1* (Solyc02g088190, a member of the *MIXTA-like* clade of Subgroup 9A) and was referred to as *SlMixta-like* by [34]. These authors silenced *SlMIXTA-like-1* in tomato and found that the transcription factor promotes the development of conical cells in the epidermis of fruit and acts as a positive regulator of fruit cuticular lipid biosynthesis and assembly. The same gene was identified by Galdon-Armero et al. [35] who used introgression lines to identify genomic regions involved in epidermal cell outgrowth. The authors used ViGS, genome editing and overexpression studies to reveal a role for *SlMIXTA-like-1* as a negative regulator of leaf trichome development and a positive regulator of petal conical cell outgrowth. Ewas et al. [36] studied the gene named in this paper *SlMIXTA-3* (Solyc01g010910, a member of the *MIXTA* clade of Subgroup 9A), which they referred to as *SlMX1* or *SlMIXTA-like-1*. This gene was shown to be involved in the modulation of drought resistance and also metabolic processes. Overexpression of *SlMIXTA-3* in tomato resulted in increased drought tolerance and improved fruit quality, while silencing by RNAi (RNA interference) resulted in the opposite [36]. The transcription factor has also been implicated in trichome initiation [36].

The diversity of epidermal cell outgrowths in tomato make it an ideal model in which to explore the function of the Myb subgroup 9 genes, with a particular focus on understanding the development of the unusual anther trichomes.

## Results

### There are seven members of the *R2R3 MYB subgroup 9* family of transcription factors in *Solanum lycopersicum*

A BLAST search of the tomato genome for the diagnostic motif of the MYB subgroup 9 transcription factor family revealed the presence of 7 candidate genes in *Solanum lycopersicum* (Fig 1A). The candidate genes were divided into the subclades 9A and 9B by the presence of a further diagnostic motif (Fig 1Aa). Subgroup membership was also confirmed by phylogenetic analysis using the phylogeny of Brockington et al., 2013 [6] as a framework. The positions of the 7 tomato genes are shown in Fig 1B and S1 Fig. Five of the genes fell into Subgroup 9A, with four of these belonging to the *MIXTA* subclade (*SlMIXTA-1*, *SlMIXTA-2*, *SlMIXTA-3*, *SlMIXTA-4*) and one belonging to the *MIXTA-like* subclade (*SlMIXTA-like-1)*. Two of the genes belonged to the *MYB17* subclade of subgroup 9B (*SlMYB17-1*, *SlMYB17-2)*. No members of the subclade *MYB17-like* were found in the tomato genome.

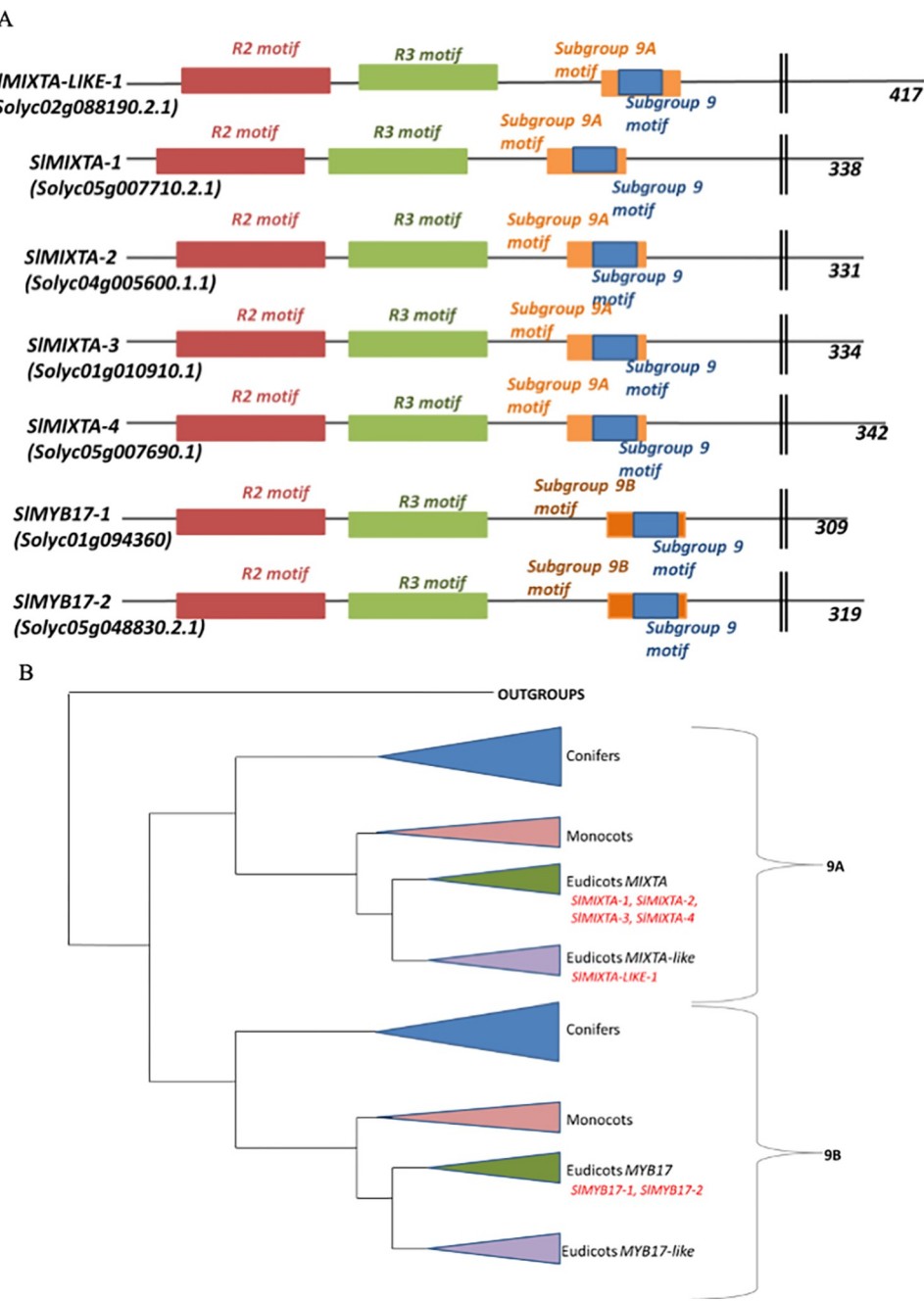

**Fig 1. A.** Cartoon of the proteins of the R2R3 MYB subgroup 9 genes of *Solanum lycopersicum*, with key motifs labelled. Sol genomics codes are displayed beneath the name the transcription factor is referred to within this paper. Lengths displayed are total amino acids. **B.** Cartoon phylogeny of the R2R3 MYB Subgroup 9 family of transcription factors. Based on phylogenetic reconstruction by Brockington et al. 2013. Placement of R2R3 subgroup 9 transcription factors within tomato is indicated.

## All of the subgroup 9A genes induced epidermal outgrowth when expressed ectopically in tobacco, but to different degrees

Five independent lines were analysed for each construct, with transgene expression confirmed by RT-PCR (S2 Fig; only 4 lines shown for *SlMIXTA-like1*). All of the transgenic plants had

generally normal growth habits, with leaves and flowers of macroscopically wild type appearance. However, microscopic analysis revealed epidermal outgrowth on a number of organs. Each of the five Subgroup 9A genes induced outgrowth of epidermal cells when ectopically expressed in tobacco. Representative SEM images of a single transgenic line per construct are shown in Fig 2, in comparison to wild type tissues (Fig 2A–2D).

Lines ectopically expressing *SlMIXTA-1* had anthers which did not dehisce even when flowers matured. This phenotype had reproductive consequences for these transgenic tobacco plants; they had to be pollinated by hand in order to produce seed. Examined using cryo-SEM the entire anther surface was covered in trichome-like epidermal cell outgrowths and conical cells (Fig 2E). Some of these outgrowths were observed to be branched, and some had stomatal guard cells at the tip of the outgrowth. The epidermal surface of the ovary, usually composed of flat cells in the wild type, had many trichome-like outgrowths which were more elongated than those found on the anther surface and were in places multi-lobed (Fig 2F). Ectopic conical cells were observed on the leaf epidermis, particularly on the adaxial side (Fig 2G1 and 2H1). Ectopic branching trichomes were observed on both the adaxial and abaxial leaf epidermis (Fig 2G2 and 2H2).

In contrast lines ectopically expressing *SlMIXTA-2* had only a weak epidermal outgrowth phenotype. The epidermal surface of the anthers had occasional glandular trichomes (Fig 2I1). Non-glandular trichome-like outgrowths were also observed along the anther connective (Fig 2I2) and on the side of the anther to a lesser degree. Anthers were able to dehisce and overall were not dramatically different from WT. The epidermal surface of the ovary was smooth and resembled WT (Fig 2J). The leaves had some branched trichomes on the adaxial side but otherwise resembled WT (Fig 2K1 and 2L1).

Lines ectopically expressing *SlMIXTA-3* also showed only a weak epidermal outgrowth phenotype and largely resembled WT. Glandular trichomes were present on the anther epidermal surface (Fig 2M1) as well as some non-glandular trichomes on the anther connective, however the anthers dehisced normally (Fig 2M). The surface of the ovary contained some cells which had grown out from the plane of the tissue, but these conical cells were not very pronounced (Fig 2N1). The leaves had branched trichomes, both glandular and non-glandular, on the adaxial surface (Fig 2O1) but the abaxial side of the leaf resembled WT (Fig 2P).

Lines ectopically expressing *SlMIXTA-4* had a stronger phenotype, similar to those expressing *SlMIXTA-1*. The general shape of the anther epidermal cells was conical (Fig 2Q1). Glandular trichomes (Fig 2Q2), non-glandular trichomes (Fig 2Q3) and stomata were also found on the anther epidermal surface (Fig 2Q4). The conical shape of the epidermal cells of the anther became more pronounced as the anther reached maturity and began to dehisce. Glandular trichomes were also more exaggerated and numerous on the anther connective. The anthers were able to dehisce, to a limited extent. Trichomes were also observed on filaments of mature anthers but not on immature anthers. The ovary epidermal cell surface had conical cells (Fig 2R1) towards the base of the ovary and some trichome like outgrowths were also observed (Fig 2R2). Branched trichomes were occasionally observed on the leaf epidermal surface on both sides of the leaf (Fig 2S and 2T).

Lines ectopically expressing *SlMIXTA-like-1* had glandular trichomes (Fig 2U1), non-glandular trichomes (Fig 2U2) and stomata (Fig 2U3) on the anther surface, and the rest of the anther was covered in ectopic conical cells (Fig 2U4). The epidermis of the ovary had conical shaped cells (Fig 2V1), however the phenotype was not as strong as seen with *SlMIXTA-1* and *SLMIXTA-4*. Occasional conical cells were also seen on the inside of the corolla tube. Branched trichomes were observed on the abaxial leaf surface (Fig 2X1).

In summary, ectopic expression of *SlMIXTA-1* and *SlMXTA-4* induced extensive epidermal cell outgrowth, *SlMIXTA-like-1* induced an intermediate phenotype, and *SlMIXTA-2* and

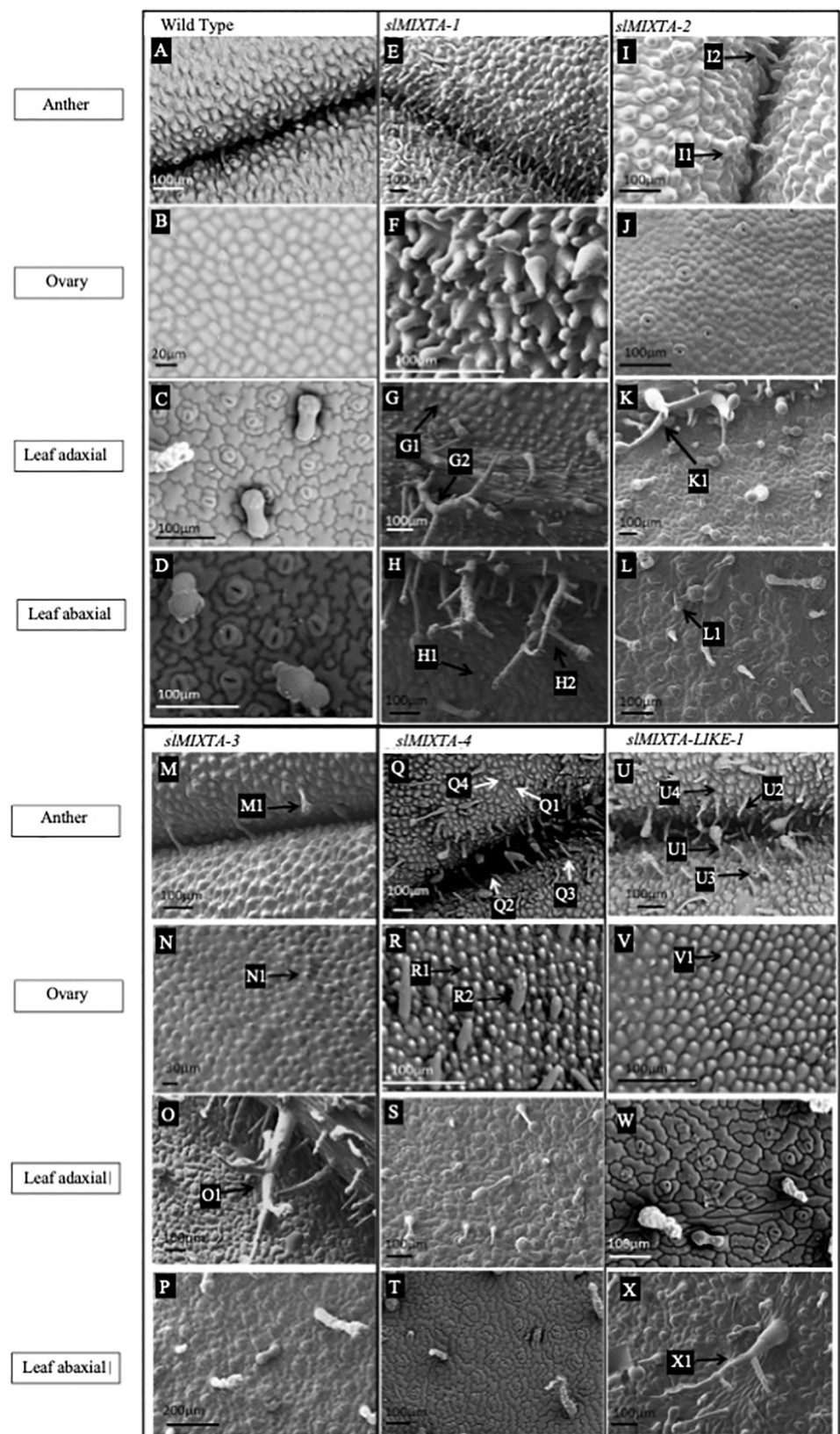

**Fig 2. SEM images of key tissues of transgenic tobacco ectopically expressing the *Solanum lycopersicum subgroup 9A* genes.** The images shown are representative individuals of each line, from which a minimum of 5 individuals were examined per line. Wild-type tobacco tissues are shown as comparison. The tissues shown are anther surface and ovary surface as these tissues showed the most distinctive transgenic phenotypes and showed the most variation with expression of the different genes. A: WT tobacco anther, B: WT ovary epidermal surface. C: WT adaxial leaf surface. D: WT abaxial leaf surface. E: Anther surface of tobacco ectopically expressing *SlMIXTA-1*. F: Ovary surface of tobacco ectopically expressing *SlMIXTA-1*. G: Adaxial leaf surface of tobacco ectopically expressing *SlMIXTA-1*, G1 labels ectopic conical cells, G2 labels ectopic branched trichome. H: Abaxial leaf surface of tobacco ectopically expressing *SlMIXTA-1*, H1 labels ectopic conical cells, H2 labels ectopic branched trichome. I: Anther surface of tobacco ectopically expressing *SlMIXTA-2*, I1 labels ectopic glandular trichome, I2 labels ectopic non-glandular trichome. J: Ovary surface of tobacco ectopically expressing *SlMIXTA-2*. K: Adaxial leaf surface of tobacco ectopically expressing *SlMIXTA-2*, K1 labels ectopic branched trichome. L: Abaxial leaf surface of tobacco ectopically expressing *SlMIXTA-2*, L1 labels ectopic branched trichome. M: Anther surface of tobacco ectopically expressing *SlMIXTA-3*, M1 labels ectopic glandular trichome. N: Ovary surface of tobacco ectopically expressing *SlMIXTA-3*, N1 labels ectopic conical cells. O: Adaxial leaf surface of tobacco ectopically expressing *SlMIXTA-3*, O1 labels ectopic branched trichome. P: Abaxial leaf surface of tobacco ectopically expressing *SlMIXTA-3*. Q: Anther surface of tobacco ectopically expressing *SlMIXTA-4*, Q1 labels ectopic conical cells, Q2 labels ectopic guard cells, Q3 labels ectopic non-glandular trichome, Q4 labels ectopic glandular trichome. R: Ovary surface of tobacco ectopically expressing *SlMIXTA-4*, R1 labels ectopic conical cells, R2 labels ectopic non-glandular trichome. S: Adaxial leaf surface of tobacco ectopically expressing *SlMIXTA-4*. T: Abaxial leaf surface of tobacco ectopically expressing *SlMIXTA-4*. U Anther surface of tobacco ectopically expressing *SlMIXTA-like-1*, U1 labels ectopic glandular trichome, U2 labels ectopic non-glandular trichome, U3 labels ectopic guard cell, U4 labels ectopic conical cell. V: Ovary surface of tobacco ectopically expressing *SlMIXTA-like-1*, V1 labels ectopic conical cell. W: Adaxial leaf surface of tobacco ectopically expressing *SlMIXTA-like-1*. X:Abaxial leaf surface of tobacco ectopically expressing *SlMIXTA-like-1*, X1 labels ectopic branched trichome.

*SlMIXTA-3* induced only weak epidermal outgrowth. Relative strength of observed phenotypes are summarised in Table 1.

## Both of the subgroup 9B genes induced extensive epidermal outgrowth when expressed ectopically in tobacco

Five independent lines were analysed for each construct, with transgene expression confirmed by RT-PCR (S3 Fig). All of the transgenic plants had generally normal growth habits, with leaves and flowers of macroscopically wild type appearance. Representative SEM images of a single transgenic line per construct are shown in Fig 3, in comparison to the same wild type tissues shown in Fig 2 (Fig 3A–3E). Both of the Subgroup 9B genes induced extensive outgrowth of epidermal cells when ectopically expressed in tobacco, producing very strong phenotypes. Relative strength of observed phenotypes are summarised in Table 2.

Lines ectopically expressing *SlMYB17-1* had a very strong anther phenotype. The anthers were unable to dehisce and the plants had to be hand pollinated. The anther epidermis was entirely converted to trichomes (Fig 3F1) and conical cells (Fig 3F2), most exaggerated at the anther connective (Fig 3F). These outgrowths were also present on the filament. Some of the

**Table 1. A table summarising the relative strengths of phenotypes of tobacco lines ectopically expressing each of the *R2R3 Myb Subgroup 9* genes from tomato.** This table summarises the relative strength of phenotypes observed when each of the 7 *R2R3 Myb Subgroup 9* genes from tomato are ectopically expressed in tobacco. Relative strength of phenotype is rated: +++ (strong), ++ (intermediate), + (weak) or 0 (no phenotype).

| | Leaf | Ovary | Anther | Overall Phenotype Strength |
|---|---|---|---|---|
| *SlMIXTA-1* | +++ | +++ | +++ | strong |
| *SlMIXTA-2* | + | 0 | + | weak |
| *SlMIXTA-3* | + | + | + | weak |
| *SlMIXTA-4* | ++ | ++ | +++ | strong |
| *SlMIXTA-like-1* | + | + | ++ | intermediate |
| *SlMYB17-1* | +++ | +++ | +++ | strong |
| *SlMYB17-2* | +++ | +++ | +++ | strong |

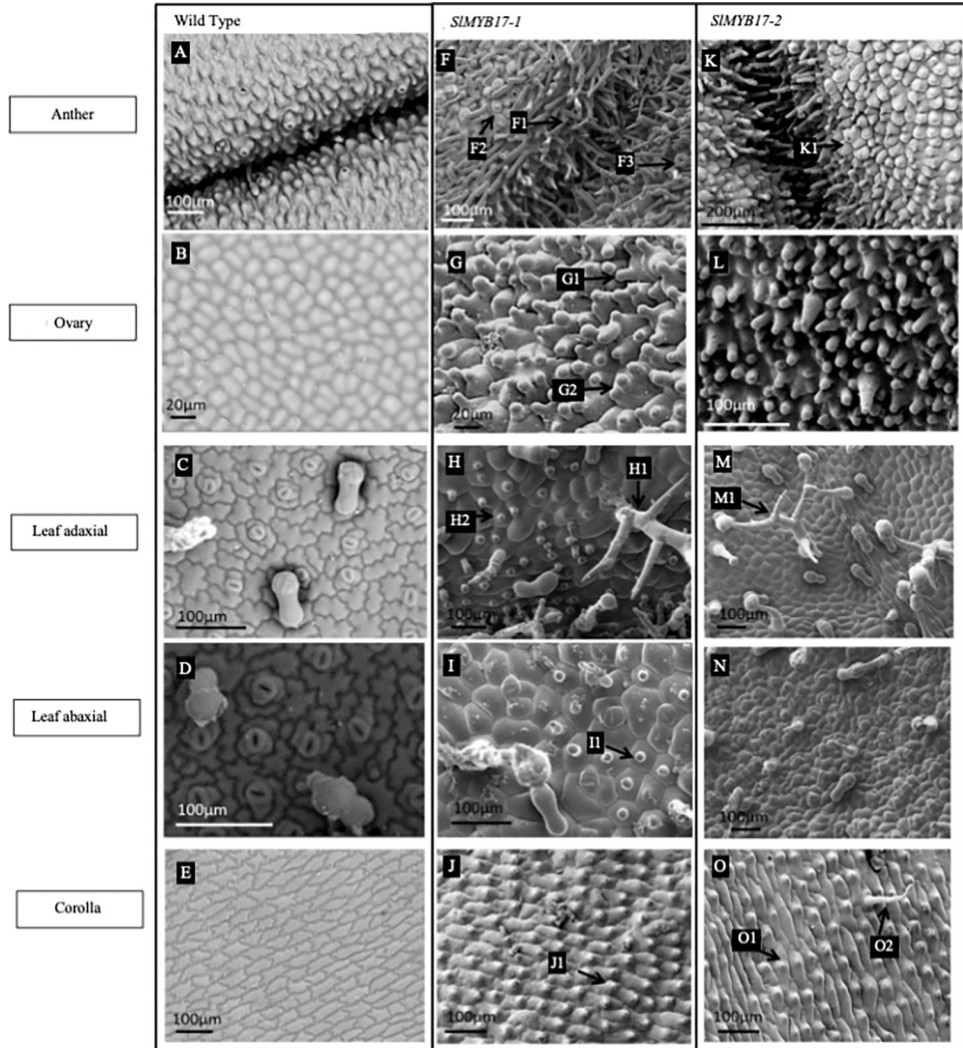

**Fig 3. The transgenic phenotypes of lines of tobacco ectopically expressing each of the *R2R3 MYB subgroup 9B genes of tomato*.** The images shown are representative individuals of each line, from which a minimum of 5 individuals were examined per line. Wild-type tobacco tissues are shown as comparison. The tissues shown are anther surface and ovary surface as these tissues showed the most distinctive transgenic phenotypes and showed the most variation with expression of the different genes. A: WT tobacco anther. B: WT ovary epidermal surface. C: WT adaxial leaf surface. D: WT abaxial leaf surface. E: WT corolla surface. F: Anther surface of tobacco ectopically expressing *SlMYB17-1*, F1 labels ectopic trichomes, F2 labels ectopic conical cells. G: Ovary surface of tobacco ectopically expressing *SlMYB17-1*, G1 labels ectopic trichome like outgrowths, G2 labels multilobed outgrowths. H: Adaxial leaf surface of tobacco ectopically expressing *SlMYB17-1*, H1 labels ectopic branched trichome, H2 labels ectopic conical cells. I: Abaxial leaf surface of tobacco ectopically expressing *SlMYB17-1*, I1 labels ectopic conical cells. J: Corolla surface of tobacco expressing *SlMYB17-1*, J1 labels ectopic conical cells. K: Anther surface of tobacco ectopically expressing *SlMYB17-2*, K1 labels ectopic guard cells on top of cell outgrowths. L: Ovary surface of tobacco ectopically expressing *SlMYB17-2*. M: Adaxial leaf surface of tobacco expressing *SlMYB17-2*, M1 labels ectopic branched trichome. N: Abaxial leaf surface of tobacco expressing *SlMYB17-2*. O: Corolla surface of tobacco expressing *SlMYB17-2*, O1 labels ectopic conical cells, O2 labels ectopic trichome.

trichomes had stomata on the tip (Fig 3F3). The epidermal surface of the ovary had trichome-like outgrowths with a mix of long and short trichomes (Fig 3G1) and some of the outgrowths were multi-lobed (Fig 3G2)). The adaxial side of the leaf had branching trichomes (Fig 3H1) and some of the epidermal surface cells had conical outgrowths (Fig 3H2). The abaxial surface

**Table 2. A table summarising the expression of the *R2R3 MYB subgroup 9* candidate genes in *Solanum lycopersicum*.** This table summarises expression positions predicted by the eFP browser for the candidate genes.

| Gene | eFP browser prediction |
|---|---|
| *SlMIXTA-1* | Unopened flower buds |
| *SlMIXTA-2* | Leaves |
| *SlMIXTA-3* | Unopened flower buds |
| *SlMIXTA-4* | Unopened flower buds and mature flowers |
| *SlMIXTA-like-1* | Unopened flower buds, leaves and fruit |
| *SlMYB17-1* | Low levels in green fruit |
| *SlMYB17-2* | Unopened flower buds, mature flowers. Lower expression throughout leaves. |

of the leaves had large numbers of branched trichomes and many conical cells (Fig 3I1). The normally flat inner corolla tube had conical cells and occasionally longer trichomes (Fig 3J1).

Lines ectopically expressing *SlMYB17-2* also had anthers which were unable to dehisce, and had to be hand pollinated. The anthers were completely covered in epidermal cell outgrowths both in the form of trichomes and conical cells (Fig 3K). Sometimes stomata were on the end of these trichomes (Fig 3K1). Longer glandular trichomes were also sometimes observed at the anther connective. The anther filaments also had trichomes on the surface. The epidermal surface of the ovary was entirely composed of cellular outgrowths which were most exaggerated at the base of the ovary (Fig 3L). Branched trichomes were seen on the adaxial leaf surface (Fig 3M1). Conical cells were seen on the abaxial surface on and around the leaf vein (Fig 3N). Some conical cells (Fig 3O1) were observed on the inside of the corolla tube along with some trichomes (Fig 3O2).

## The anther trichome mesh develops at an intermediate stage of bud development

To assess the timing of anther trichome development we divided tomato flower development into 6 stages, mainly determined by the relative position of the calyx and the corolla (Fig 4A). Scanning electron microscopy revealed that the anther trichomes appear first as outgrowths in stage 2 (Fig 4B2I), expanding in stage 3 and knitting the anthers together by late stage 3 (Fig 4B3II). Later in flower development additional cellular outgrowth is observed on the anther epidermis, with multilobed cells appearing at stage 4 (Fig 4B4III).

## Several of the *subgroup 9 Myb* genes are expressed during early stages of tomato flower development

The tomato eFP browser http://bar.utoronto.ca/efp_tomato/cgi-bin/efpWeb.cgi) gave an indication of which tissues the candidate genes were expressed in, and of the levels of expression. These data are summarised in Table 2. Since a number of the genes appear to be expressed during flower development, we used semiquantitative RT-PCR to explore the timing of expression in floral tissues relative to the development of the anther trichome mesh (Fig 5).

Expression was explored in whole floral bud at stage 1 (before anther trichomes emerge) and stage 2 (the earliest stage of anther trichome development), and in dissected anthers at stage 3 (when the trichome mesh is knitting together) and stage 4.

*SlMIXTA-1* was expressed in the early stages of the bud development. Bands were visible in stage 1 and stage 2, most strongly in stage 1 with the expression fading as the bud developed: only a very faint band was visible in stage 4 at cycle 40. *SlMIXTA-2* was not expressed in the

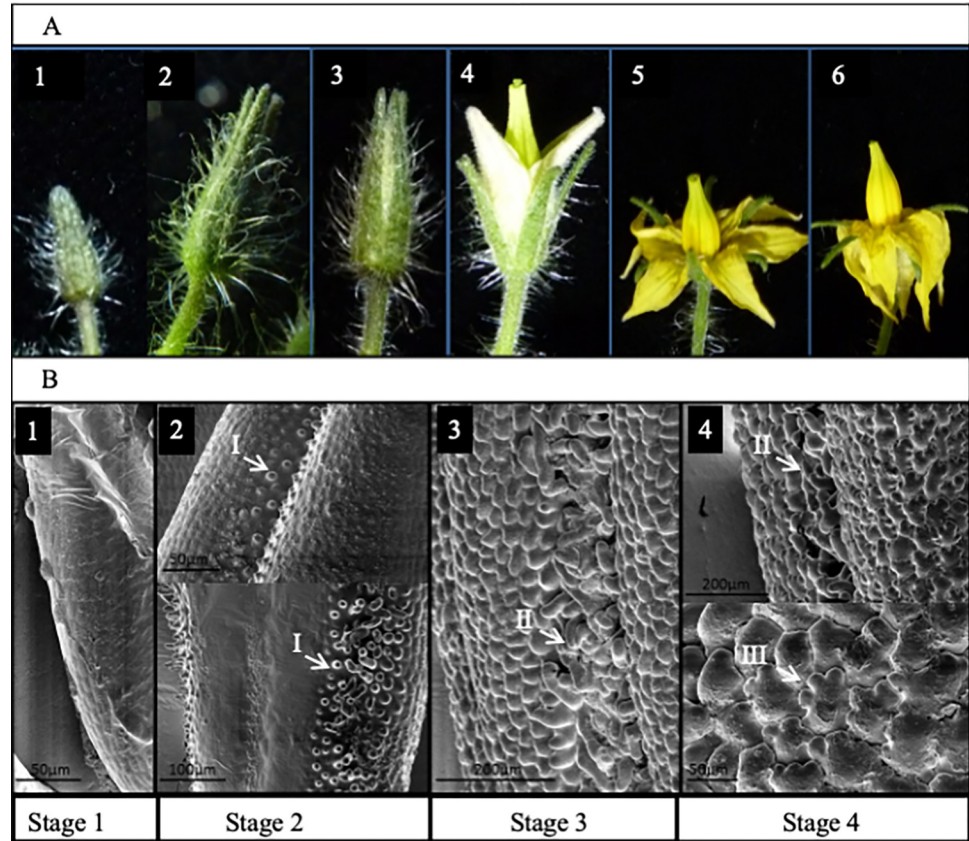

**Fig 4. The stages of floral development and the anther epidermal cell outgrowths of *Solanum lycopersicum*.** A. Six stages of tomato flower development chosen to reflect the development of the epidermal cell outgrowths on tomato anthers. B. SEM images of anther surface at stages 1 to 4. Epidermal cell outgrowths begin to develop at stage 2 (I). By stage 3 the anther trichome mesh (II) is almost completely formed and other papillae have begun to form on the anther surface. The papillae on the anther surface take on a distinctive 'glove-like' multilobed (III) appearance during stage 4.

floral tissues examined, or expressed at such a low level that it was not detectable, as predicted by the tomato eFP browser. *SlMIXTA-3* was expressed only at a low level in all four of the floral stages. *SlMIXTA-4* also had only very low level expression visible only after 40 cycles for each of the tissues. *SlMIXTA-like-1* appeared to not be expressed in the floral stages examined, or only expressed at low levels. A band was visible only at cycle 40 and only very faintly in all tissues examined (a little stronger in bud stage 2). The *SlMYB17-1* gene was expressed most strongly in the early stages of the development of the bud and during the stages in which the anther trichome mesh was developing. The gene was expressed in the floral tissue at stages 1 and 2 (especially strongly compared to the reference gene in stage 2), after which expression level dropped. By stage 4 the band was very faint and only visible after 40 cycles. The *SlMYB17-2* gene was expressed in all floral stages studied, but was expressed most strongly in early stages of bud development: the brightest bands were observed in stage 1 and especially 2 compared with the housekeeping gene. This was an almost identical expression pattern to that observed for *SlMYB17-1*.

Since stage 1 and 2 samples contained entire bud tissue, while stage 3 and 4 samples only contained anthers, any expression observed in stages 1 and 2 may not only arise from anther trichome development but might also reflect activity in the petals or sepals.

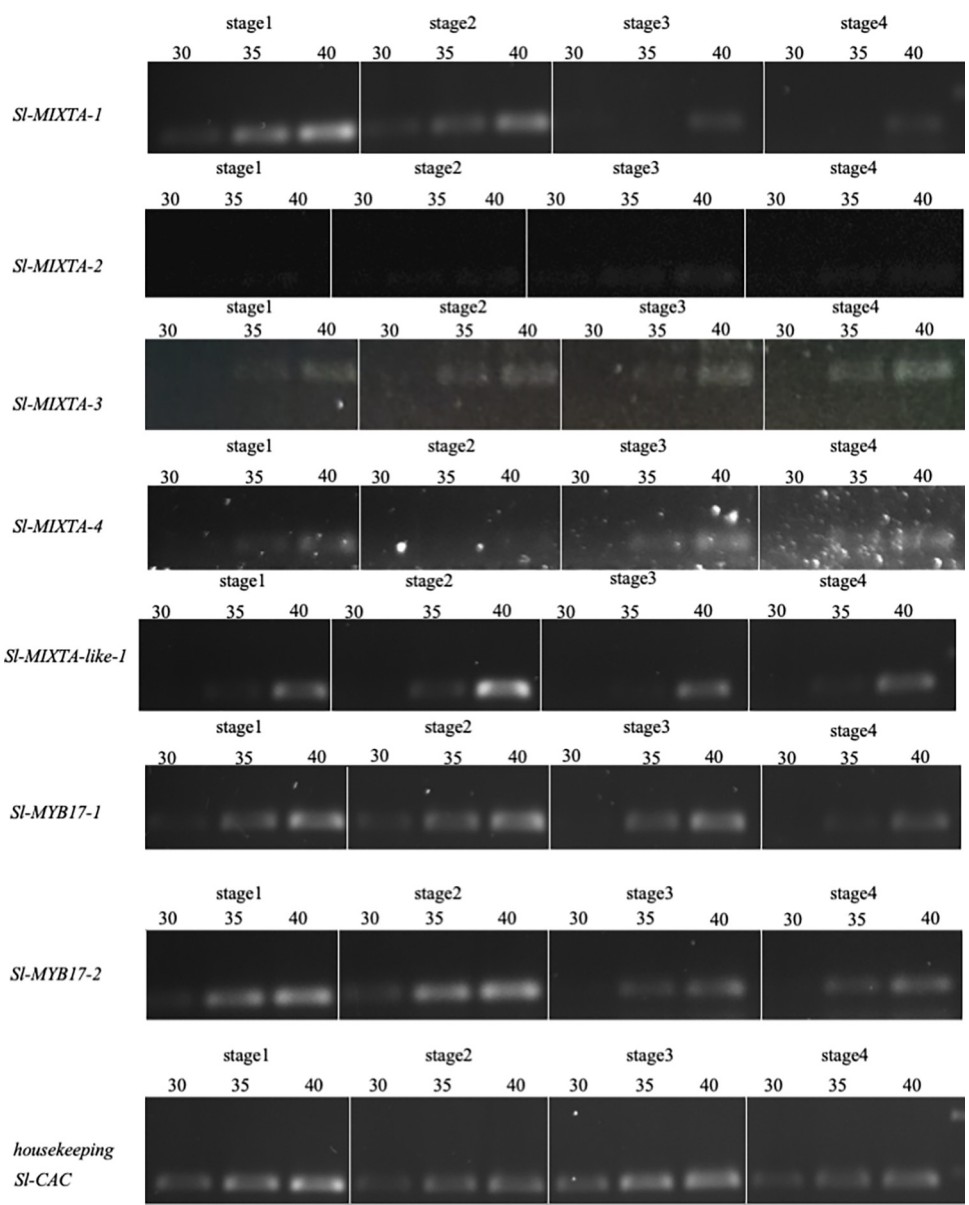

**Fig 5. Semi quantitative RT PCR analysis of expression of all *Solanum lycopersicum R2R3 MYB subgroup 9* genes during development of the flower.** Stages 1 to 4 are the first 4 developmental stages shown in Fig 4. Flower developmental stages are labelled above each lane, and the number of cycles is indicated. Positive and negative controls were conducted for each primer set (not shown): the positive control was the same primers amplifying from a plasmid containing the sequenced gene, the negative control was the same primers with only water. *SlCAC* was used as a reference gene (lower panels).

## Discussion

The R2R3 Myb subgroup 9 family of transcription factors of *Solanum lycopersicum* were shown to various degrees to be capable of inducing outgrowth of cells when they were ectopically expressed in tobacco. This indicates that all these proteins have the potential to initiate epidermal cell outgrowths such as conical cells and trichomes within *Solanum lycopersicum*. However, the degree to which epidermal cell outgrowths were induced, and the number of tissues in which they were observed to act, varied from gene to gene.

The range of phenotypes exhibited in this study was similar to that reported in the set of studies in which the *R2R3 subgroup 9A* genes of *Antirrhinum majus (A. majus)* were ectopically expressed in tobacco [8, 10, 37, 38]. The strongest phenotype in those previous studies was observed in tobacco lines containing ectopically expressed *AmMIXTA* [8]. In these lines trichomes were observed covering most tissues and with a particularly large amount of epidermal cell outgrowth observed on the ovary. These outgrowths included branched and glandular trichomes on the ovary and the production of conical cell protrusions on the epidermis on both sides of the leaf. None of the *R2R3 subgroup 9* genes of *Solanum lycopersicum* produced phenotypes quite as extreme as this when ectopically expressed in tobacco. The strongest phenotypes were those seen in lines expressing the two *subgroup 9B* genes *SlMyb17-1* and *SlMyb17-2*. These phenotypes were reminiscent of that of tobacco expressing *AmMIXTA* (a *subgroup 9A* gene). The majority of tissues exhibited epidermal cell outgrowths not found in the WT, with the ovary and the anthers particularly covered in trichomes of various types. However, no branched trichomes were found on the surface of the ovary, although the trichomes also sometimes had stomata on the end. Our sqRT-PCR analysed revealed that these two genes had nearly identical expression patterns in the tomato flower, with particularly strong expression at early stages of development, when the trichome mesh is beginning to form. This contradicts the eFP browser, which predicted that *SlMyb17-1* would not be expressed in the flower but that *SlMyb17-2* would.

The outgrowths on the ovary of the transgenic lines expressing *SlMyb17-1* resembled the 'glove-like' papillae found on the surface of the tomato anther (Fig 4B). Meanwhile, the outgrowths on the anther of these lines were very like the trichomes which make up the trichome mesh of WT tomato anthers. In combination with its expression profile in developing flowers (and persisting in later stages of anther development) this gene is a good candidate for the control of the development of the trichome mesh and/or 'glove-like' papillae on the anther surface. The transgenic lines expressing *SlMyb17-2* also developed trichomes on the anthers and ovary that resembled those of the trichome mesh. The two *SlMyb-17* genes could be considered likely candidates for the control of the development of the trichome mesh. They were both expressed most strongly in tissue stages 1 and 2, where the trichome mesh is developing. It is possible that the two genes function together or are redundant with one another. Previously studied members of the *MYB17* clade of genes in subgroup 9B have not shown an involvement in epidermal cell outgrowth. *AtMYB17* [39] has been shown to be involved in flowering commitment, but no epidermal phenotype was seen in a mutant line. The gene was also shown to be involved in the regulation of activity of *APETALA1* in the flowers of *Arabidopsis thaliana* and is thought to act together with *LEAFY* [40]. However it has been previously argued that with so much paralogy in the *MYB subgroup 9* lineages it is possible that the *AtMYB17* gene of *Arabidopsis thaliana* may have acquired a different role to other *MYB17* representatives and that a possible role of *MYB17* genes in the regulation of epidermal cell outgrowths should not be dismissed entirely [6]. Brockington et al. [6] also implicated the *MYB17* lineage genes in epidermal outgrowth regulation because the *Nicotiana* EST-derived fragments nested within the *MYB17* clade in their phylogenetic analysis were derived from trichome-specific transcriptomes.

A *MYB17-like* gene (*LjMYB17-like)* from *Lotus japonicus* was analysed by Brockington et al. [6] and, when ectopically expressed in tobacco, produced a very strong phenotypic effect. The epidermal cells on the adaxial and abaxial leaf surfaces became conical in shape and there was a reduced number of stomata. The filament of the stamen also gained trichomes and conical cells on its epidermal surface. The ovary surface was covered in long conical cells and the cells on the petal lobes had become elongated and glandular trichomes were also found [6]. No

member of the *MYB17-like* clade was present in the tomato genome, but this previous study supports a role for *subgroup 9B* genes more generally in epidermal cell outgrowth.

The *A. majus* subgroup 9 gene with the second strongest phenotype when expressed in tobacco was *AmMYBML-1*, another gene in the *MIXTA* clade [37]. The anthers of these lines were covered in conical cells and failed to dehisce as a result. The ovary surface was covered in a mixture of conical cells and trichomes. However the epidermal surface of the leaves was considered to be WT in appearance. The phenotype of lines expressing *SlMIXTA-1* was reminiscent of this and could be considered the third strongest phenotype observed in this study. A mixture of conical cells and trichomes were observed on the ovary surface and trichomes were also found on the anther surface and consequently the anthers did not dehisce. Branched trichomes were also observed on the leaf surface and occasional conical protrusions on the leaf surface, however in general the phenotype was weaker than that observed in *AmMIXTA* lines and closer in resemblance to those of *AmMYBML-1*.

The *SlMIXTA-4* lines were also reminiscent in phenotype of the *AmMYBML-1* study. However the phenotype was less strong than observed in *SlMIXTA-1* lines. The ovary still exhibited both conical cells and trichomes, but the proportion of conical cells relative to trichomes was increased. The number of trichome outgrowths observed on the anther surface was less than that observed in *SlMIXTA-1* lines and the anthers were able to dehisce as a result of only slightly conical shaped cells and some glandular trichomes rather than large numbers of simple trichome-like outgrowths. The leaves resembled WT tobacco.

Both the *SlMIXTA-1* and the *SlMIXTA-4* genes were expressed in developing flowers, but *SlMIXTA-1* was expressed slightly more strongly in the early stages of development, while *SlMIXTA-4* was expressed more strongly in later stages. This temporal separation of expression patterns could indicate differential roles in flower development, with *SlMIXTA-4* potentially involved in later stage developmental processes such as the development of the glove-like papillae on the anthers.

The weakest phenotypes observed in this were those produced by ectopically expressing *SlMIXTA-2* and *SlMIXTA-3*, which resulted in an even weaker phenotype than the weakest of the phenotypes obtained from expression of *A. majus* genes. Lines expressing *SlMIXTA-2* and *Sl-MIXTA-3* had a few branched trichomes on the leaf epidermis, and a few shallow conical cells on the ovary surface, similar to those found when expressing *AmMYBML3* [38]. The lack of expression of *SlMIXTA-2* and *Sl-MIXTA-3* in floral tissues suggests that these genes do not play a role in anther trichome mesh regulation. In a previous study of *SlMIXTA-3* (*SlMX1*) it was shown that when over-expressed in tomato there was increased resistance to drought [36]. SEM images in that study showed increased numbers of trichomes on the leaves and stems of tomato lines overexpressing *SlMIXTA-3* as well as increased leaf thickness. RNAi lines with downregulation of *SlMIXTA-3* expression showed the opposite [36].

The *A.majus* genes belonging to the *MIXTA-like* clade of Brockington et al. [6] (*AmMYBML2* and *AmMYBML3*) had the weakest phenotypes when expressed in tobacco [10, 38]). The *MIXTA-like* genes from *Arabidopsis thaliana* (*AtMYB16*) and *Petunia hybrida* (*PhMYB1*) produced near identical phenotypes to that produced by ectopic expression of *AmMYBML2* [10], with some conical cells on the ovary epidermis and an extension of the petal conical cells. The *TtMYBML2* gene of *Thalictrum thalictroides* also induces conical cells on the ovary and carpel and elongates those of the petal lobe [11]. The lines expressing *SlMIXTA-like-1* in this study were reminiscent of this phenotype, yet slightly stronger. The ovary surface exhibited only conical cells and no trichomes, like the *AmMYBML2* and *AmMYBML3* phenotypes, yet conical cell-like protrusions were also observed on the anther surface (although they did not affect dehiscence). The conical cells on the petal lobe, where conical cells are observed in WT tobacco, also appeared longer in the *SlMIXTA-like-1*

expressing lines. The *SlMIXTA-like-1* gene was found not to be expressed in flowers, suggesting no role in anther trichome development. *SlMIXTA-like-1* has been previously studied and shown to be expressed significantly during tomato fruit development in both skin and flesh tissues ([34], where it was referred to as *SlMixta-like*). It was also shown that RNAi lines in which the gene was silenced in tomato resulted in the flattening of epidermal cells and thinning of the cuticle in tomato fruit [34], so it is possible that the gene is involved in epidermal cell outgrowths in other surfaces in addition to fruit cuticle. *SlMIXTA-like-1* has since been shown using CRISPR Cas9 knockout to be a negative regulator of trichome development in leaves [35] but to be a positive regulator of conical cell outgrowths in petals and fruit, therefore serving different outgrowth regulatory purposes in different tissues [35].

Transgenic experiments using a heterologous host must always be interpreted with caution. In this study we have not demonstrated that a particular gene performs a particular function, because we have not worked in the endogenous host. However, by expressing the 7 members of this subfamily from the same promoter in tobacco under the same conditions we can draw conclusions about the relative ability of each protein to induce cellular outgrowth.

This study presents the first analysis of the complete set of MYB subgroup 9 transcription factors in a single species. The cellular outgrowth induced by members of this transcription factor family plays important roles in various aspects of plant development and physiology, including in interactions with herbivores and pollinators and in macroscopic shaping of organs. Further analyses of the regulation of cellular outgrowth will allow greater understanding of these processes and support potential applications in improving crop plants.

## Materials and methods

### RNA extraction and cDNA synthesis

Wild Type (WT) *Solanum lycopersicum* was the cultivar 'Moneymaker'. For isolation of R2R3 Myb subgroup 9 candidate genes, tissue was harvested from flowers, young leaves, buds of various floral growth stages, cotyledons, young roots, hypocotyls and apical meristems. These tissues were pooled. Tissue selection was guided by use of the Tomato efp Browser at Bar. UToronto.ca, Rose Lab Atlas (http://bar.utoronto.ca/efp_tomato/cgi-bin/efpWeb.cgi). Concert Plant™ RNA Reagent (Invitrogen) was used as per manufacturer's instructions. RNA was DNase treated and purified using phenol:chloroform purification. cDNA was synthesised using BioScript™ (Bioline). RNA for semi-qRTPCR was extracted using a Trizol buffer method. The cDNA for semi-qRTPCR was synthesised using the Superscript II retrotranscription Kit (Invitrogen).

### Identification and isolation of all members of R2R3 MYB Subgroup 9 family of transcription factors from *Solanum lycopersicum*

All members of the R2R3 MYB subgroup 9 family of transcription factors in *Solanum lycopersicum* were identified through a BLAST search (NCBI) for the conserved motif of the subgroup 9A and 9B transcription factor families. The subgroup membership was also confirmed by phylogenetic analysis using the phylogeny of [6] as a framework. This phylogeny was a GARL1 maximum likelihood phylogram of 220 members of the subgroup 9 R2R3 Myb genes and the candidate genes were manually aligned with the phylogram at the level of amino acids. ML bootstrap analysis was conducted using default parameters and 100 replicates. Five replicates for GARL1 analysis were conducted and the topology with the highest likelihood score selected. In this study the phylogenetic analysis was conducted on nucleotide alignments which were aligned previously by codon; any non-confident aligned amino acids were

excluded during re-iterative preliminary analysis which explored effects of exclusion and the most robust phylogenetic trees selected from preliminary analysis.

Gene specific primers were designed to amplify the full length coding sequence of each of the candidate genes. Primers used can be found in Supplementary Information (S1 Table). Coding sequences were amplified using Phusion High Fidelity DNA Polymerase. Correct amplification was confirmed by sequencing with gene specific primers and alignment using Clustal Omega against the sequenced tomato genome as viewed on Phytozome V12.1 and Sol Genomics Network (Current Tomato Genome Version SL3.0 and Annotation ITAG3.10).

## Tobacco transformation

The coding sequence of each of the R2R3 subgroup 9 genes was cloned into a modified version of pGreen [41], containing two copies of the CaMV35s promoter and the 35S terminator to drive constitutive expression in plant tissues.

*Nicotiana tabacum* 'Samsun' was transformed using a modification of the leaf disk method of [42]. Tobacco was selected as the host for ectopic expression because of its status as a model species for transgenic studies due to its ease of transformation and its well characterised phenotype. It has also been used for ectopic expression in many previous studies exploring the function of MYB subgroup 9 proteins. Therefore, this allows functional comparison between genes from different species.

## Genotyping of transgenic tobacco

PCR with genomic DNA as template used gene specific primers or a gene specific forward primer with the 35S Reverse Primer (detailed in Supplementary Information). Once presence of the transgene was confirmed, expression was analysed. RNA was extracted using either a CTAB-based protocol or Trizol buffer, and was cleaned using a phenol:choloroform purification before DNase treatment. PCR with RNA template was used to confirm the absence of gDNA before cDNA synthesis using the Superscript II retrotranscription Kit (Invitrogen). PCR to confirm expression of the transgene was conducted using gene specific primers, with ubiquitin primers as a positive control and WT tobacco gDNA as a negative control.

## Phenotyping of transgenic tobacco

A minimum of five transgenic lines per construct, all shown to be expressing the transgene, were analysed.

Characterisation of transgenic line phenotypes was conducted using the Keyence light microscope VHX-5000 and the Zeiss EVO HD15 cryo-scanning Electron Microscope. For SEM tissue was mounted using a mix of colloidal graphite (G303, Agar Scientific.ltd. unit 7) and O.C.T compound (Scigen Tissue-Plus®, O.C.T. Scigen Scientific Gardena, LA90248USA). This glue was mixed in a ratio of 1/3 colloidal graphite to 2/3 O.C.T. The samples were cryogenically frozen and then underwent a sublimation of 5–9 minutes at -90˚C. They were sputter coated with 5nm of platinum.

## Gene expression analysis

Floral stages for gene expression analysis were determined according to macroscopic features of organ position and trichome mesh development on the anthers, defined at each stage. Stages of tomato flower development were imaged using SEM of epoxy resin casts [43]. Three pools of tissue were collected for each stage. Each pool contained multiple individuals, but approximately the same number of individuals were in each pool. For Stage 1 and Stage 2 the whole

bud was collected. For later stages anthers were dissected separately. The reference gene used for the semi-qRTPCR was the tomato *CAC* gene (SGN-U314153, Clathrin adaptor complex Subunit), which has previously been shown to have consistent and stable expression levels in tomato floral tissue [44]. The reference gene primers used were those described in [44]. *Myb* gene primers were tested for specificity against each of the 7 genes cloned into pBLUEscript. 5µl of the PCR reaction was removed after 30, 35 and 40 cycles and analysed on a 1.5% agarose gel.

## Supporting information

**S1 Fig. Phylogenetic analysis of R2R3 MYB subgroup 9 genes (with the position of tomato genes included).** S1A shows subgroup 9A. S1B shows subgroup 9B.
(PDF)

**S2 Fig. Analysis of expression of *subgroup 9A* genes in transgenic tobacco lines.**
(PDF)

**S3 Fig. Analysis of expression of *subgroup 9B* genes in transgenic tobacco lines.**
(PDF)

**S4 Fig. Uncropped gel images used for Fig 5, S2 and S3 Figs.**
(PDF)

**S1 Table. Table of primers used.**
(PDF)

## Acknowledgments

We thank Matthew Dorling for excellent care of plants, Sam Brockington for help with phylogenetic analysis, and members of the Glover lab for helpful discussions.

## Author Contributions

**Supervision:** Beverley J. Glover.

**Writing – original draft:** Gwendolyn V. Davis.

**Writing – review & editing:** Beverley J. Glover.

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
