## [Decision Letter · Decision Letter 0]

25 Sep 2023

PONE-D-23-25238Characterisation Of The R2R3 Myb Subgroup 9 Family Of Transcription Factors In Tomato

PLOS ONE

Dear Dr. Davis,

Thank you for submitting your manuscript to PLOS ONE. After careful consideration, we feel that it has merit but does not fully meet PLOS ONE’s publication criteria as it currently stands. Therefore, we invite you to submit a revised version of the manuscript that addresses the points raised during the review process.

We look forward to receiving your revised manuscript.

Kind regards,

Nguyen Hoai Nguyen

Academic Editor

PLOS ONE

Journal Requirements:

2. Please note that in order to use the direct billing option the corresponding author must be affiliated with the chosen institute. Please either amend your manuscript to change the affiliation or corresponding author, or email us at plosone@plos.org with a request to remove this option.

Reviewers' comments:

Reviewer's Responses to Questions

**Comments to the Author**

1. Is the manuscript technically sound, and do the data support the conclusions?

Reviewer #1: Yes

Reviewer #2: Yes

2. Has the statistical analysis been performed appropriately and rigorously? 

Reviewer #1: Yes

Reviewer #2: Yes

3. Have the authors made all data underlying the findings in their manuscript fully available?

Reviewer #1: Yes

Reviewer #2: Yes

4. Is the manuscript presented in an intelligible fashion and written in standard English?

Reviewer #1: Yes

Reviewer #2: Yes

5. Review Comments to the Author

Reviewer #1: The paper effectively introduces a novel research focus: a analysis of the R2R3 Myb Subgroup 9 transcription factor family in tomato's epidermal cell outgrowths. It offers a strong background and underscores the topic's significance. The methodology overview is concise but would benefit from a brief rationale for selecting tobacco as the host for ectopic expression. While the content mentions variation in epidermal cell outgrowth, specific phenotypic data for each ectopically expressed gene would strengthen the presentation. Identifying SlMIXTA-2 and SlMYB17-2 as candidate regulators is promising, but clarifying the selection criteria is advisable. Expanding on the implications of these findings for plant biology and applications would provide a broader context.

Here are some comments and questions for the authors.

1. Identifying and isolating all R2R3 MYB Subgroup 9 family members in Solanum lycopersicum follows a well-documented and standard procedure. This process combines BLAST searches, phylogenetic analysis, and gene-specific primer design to ensure accurate gene identification. However, a few points should be considered for further clarification:

- Phylogenetic Analysis: While the paper mentions the use of a phylogeny from a previous study (GARL1 maximum likelihood phylogram of 220 members of the subgroup 9 R2R3 Myb genes), it would be beneficial to provide more details about the specific methodology and parameters used in generating this phylogeny, as the accuracy of subgroup assignment relies heavily on this analysis.

- The paper mentions the use of the sequenced tomato genome as viewed on Phytozome V12.1 and Sol Genomics Network, with the current tomato genome version SL3.0 and Annotation ITAG3.10. Given that genome annotations can evolve over time, did you encounter any discrepancies or challenges in aligning candidate genes with the reference genome annotations? How were such issues resolved?

2. The authors present compelling evidence that both Subgroup 9B genes induce extensive epidermal cell outgrowths when ectopically expressed in tobacco. The detailed SEM images and phenotypic descriptions provide valuable insights into the effects of these genes on tobacco plants. Please clearify these contents:

- Tissue-Specific Effects: Are there tissue-specific effects of these genes' expression in tobacco? Do certain tissues or organs exhibit more pronounced epidermal cell outgrowths, and is there a correlation with their native functions in Solanum lycopersicum?

- Stomata Formation: The presence of stomata on some trichomes is intriguing. Could you discuss the potential implications of this observation? Is there evidence to suggest a functional role for these stomata in the context of altered epidermal structures?

- Reproductive Consequences: The inability of anthers to dehisce in some transgenic lines is notable. Have you explored the reproductive consequences of this phenotype? Does it impact pollination or seed production in tobacco plants, and could similar effects be expected in Solanum lycopersicum?

- Cellular Changes in Ovary: For the elongated and multi-lobed trichome-like outgrowths on the ovary surface, have you conducted any cellular or histological studies to examine the structural changes at the cellular level? Are there indications of altered cell expansion or differentiation patterns?

Reviewer #2: This manuscript are not really careful preparation. Many words are not same format, especially figures. Also Table is attached by image not by Table tool. I already mentioned all my commendations in file and authors need to recheck them.

6. PLOS authors have the option to publish the peer review history of their article (what does this mean?). If published, this will include your full peer review and any attached files.

Reviewer #1: No

Reviewer #2: No

---

## [Author Response · Author response to Decision Letter 0]

19 Oct 2023

Dear Reviewers, 

Thank you for your time in reviewing this manuscript for submission to PLOS ONE. Please find bellow listed responses to the Reviewers' comments; 

Reviewer #1: 

1. The methodology overview is concise but would benefit from a brief rationale for selecting tobacco as the host for ectopic expression. 

Response: The following explanation has been added into methods section: “Tobacco was selected as the host for ectopic expression because of its status as a model species for transgenic studies due to its ease of transformation and its well characterised phenotype. It has also been used for ectopic expression in many previous studies exploring the function of MYB subgroup 9 proteins. Therefore, this allows functional comparison between genes from different species.” 

2. While the content mentions variation in epidermal cell outgrowth, specific phenotypic data for each ectopically expressed gene would strengthen the presentation.

Response: Thank you for this helpful suggestion, we have included a table which summarises the phenotypes induced by all 7 genes in 3 major tissues.

3. Identifying SlMIXTA-2 and SlMYB17-2 as candidate regulators is promising, but clarifying the selection criteria is advisable. 

Response: Additional sentences have been added to the paper’s conclusion expanding on this slightly: “SlMIXTA-2 is expressed most strongly in the leaves, and not expressed in the floral tissue. SlMYB17-2 is the best candidate for the regulation of the anther trichome mesh. SlMYB17-2 is expressed strongly in the floral tissue and produces a clear phenotype of epidermal cell outgrowths when ectopically expressed in tobacco.”

4. Expanding on the implications of these findings for plant biology and applications would provide a broader context. 

Response: Thank you for this helpful suggestion. We have now added a new final paragraph addressing this:

“The cellular outgrowth induced by members of this transcription factor family plays important roles in various aspects of plant development and physiology, including in interactions with herbivores and pollinators and in macroscopic shaping of organs. Further analyses of the regulation of cellular outgrowth will allow greater understanding of these processes and support potential applications in improving crop plants.”

5. - Phylogenetic Analysis: While the paper mentions the use of a phylogeny from a previous study (GARL1 maximum likelihood phylogram of 220 members of the subgroup 9 R2R3 Myb genes), it would be beneficial to provide more details about the specific methodology and parameters used in generating this phylogeny, as the accuracy of subgroup assignment relies heavily on this analysis. 

response: Following has been added to the relevant part of the methods section, further expanding upon the phylogenetic analysis conducted in (Brockington et al, 2013) “ML bootstrap analysis was conducted using default parameters and 100 replicates. Five replicates for GARL1 analysis were conducted and the topology with the highest likelihood score selected. In this study the phylogenetic analysis was conducted on nucleotide alignments which were aligned previously by codon; any non-confident aligned amino acids were excluded during re-iterative preliminary analysis which explored effects of exclusion and the most robust phylogenetic trees selected from preliminary analysis.” 

6. The paper mentions the use of the sequenced tomato genome as viewed on Phytozome V12.1 and Sol Genomics Network, with the current tomato genome version SL3.0 and Annotation ITAG3.10. Given that genome annotations can evolve over time, did you encounter any discrepancies or challenges in aligning candidate genes with the reference genome annotations? How were such issues resolved?

Response: We did not encounter any difficulties. Repeated checks of the genome assemblies only ever revealed these 7 genes.

7. Clarify: Tissue-Specific Effects: Are there tissue-specific effects of these genes' expression in tobacco? Do certain tissues or organs exhibit more pronounced epidermal cell outgrowths, and is there a correlation with their native functions in Solanum lycopersicum? 

Response: In response to this we have added a new table summarising these tissue specific effects for each gene.

8. clarify: Stomata Formation: The presence of stomata on some trichomes is intriguing. Could you discuss the potential implications of this observation? Is there evidence to suggest a functional role for these stomata in the context of altered epidermal structures? 

Response: There is some evidence to suggest (at least in Arabidopsis) that the regulatory circuits of stomata, trichomes and other cell type patterning in various tissues are interlinked. However the nature of this developmental link is not fully understood (Simon et al, Plant Direct 2020; Torrii Ann Bot 2021). The presence of stomata on trichomes is likely a consequence of the over-expression of these trichome related genes interacting with the regulatory circuit of stomata formation/patterning. There is probably no functional role for these stomata; or at least none that we have evidence for. It would be fascinating if future studies uncovered the link between stomata and trichome development and cell type pattern formation. 

9. Reproductive Consequences: The inability of anthers to dehisce in some transgenic lines is notable. Have you explored the reproductive consequences of this phenotype? Does it impact pollination or seed production in tobacco plants, and could similar effects be expected in Solanum lycopersicum? 

Response: In the paper it is already mentioned that these plants had to be hand pollinated. We have expanded this sentence to provide greater clarity as to the reproductive consequences, as follows: “Lines ectopically expressing SlMIXTA-1 had anthers which did not dehisce even when flowers matured. This phenotype had reproductive consequences for these transgenic tobacco plants; they had to be pollinated by hand in order to produce seed.”

Therefore the reproductive consequences of this phenotype are apparent (and quite stark.) We note that ectopic expression of related genes from Antirrhinum majus also prevented tobacco anther dehiscence (Glover et al 1998, Development) and we believe that the phenomenon is a result of conversion of those cells usually responsible for anther dehiscence into conical or trichome cell fates.

The anthers of tobacco and tomato are very different: tobacco anthers dehisce by opening up to release pollen, whilst tomato anthers do not. Tomato (along with all members of the genus Solanum) has poricidal anthers: pollen contained within a tube-like anther is released from pores at the tip in response to buzz pollination. As a result of this difference of anther type and pollination mechanism such outgrowth on the anther surface would not have the same effect in tomato: there is no dehiscence by opening to be prevented. In fact, cell outgrowths on the surface of tomato anthers are likely possible for this very reason: they do not need to dehisce so there is no interference in the anthers’ ability to release pollen. It is possible that such outgrowths are in fact beneficial for pollinator grip during buzz pollination (like conical cells on flower petals). 

10. Cellular Changes in Ovary: For the elongated and multi-lobed trichome-like outgrowths on the ovary surface, have you conducted any cellular or histological studies to examine the structural changes at the cellular level? Are there indications of altered cell expansion or differentiation patterns? 

Response: We have not conducted any studies on the ovary surface, beyond the SEMs shown here. Future studies will analyse the ectopic cell outgrowths on each tissue and arising from each gene systematically, but that is beyond the scope of this manuscript. 

Reviewer #2

11. Many words are not same format, especially figures. 

Response: Thank you for noting this. It has now been fixed. 

12. Table is attached by image not by Table tool. 

Response: Thank you for this feedback, the appropriate tool has now been used for the table. 

13. I already mentioned all my commendations in file and authors need to recheck them. 

Response: Thank you for your feedback; all your proposed changes have been implemented. 

14. Fig. 1A and 1B should be consolidated into one file. Fig. 1 can be just one file, comprising panels A to B, rather than one file for panels A and one file for panel B. 

Response: These have now been consolidated into a single file for submission. 

Dear Nguyen Hoai Nguyen, Academic Editor at PLOS ONE, 

Thank you for your time in reviewing this paper. I have addressed your notes on the additional journal requirements as follows: 

Journal Requirements:

We have checked and formatted to adhere to these style requirements. 

2. Please note that in order to use the direct billing option the corresponding author must be affiliated with the chosen institute. Please either amend your manuscript to change the affiliation or corresponding author, or email us at plosone@plos.org with a request to remove this option. 

We have changed the corresponding author to Beverley J. Glover to adhere to this point. 

3. PLOS ONE now requires that authors provide the original uncropped and unadjusted images underlying all blot or gel results reported in a submission’s figures or Supporting Information files. This policy and the journal’s other requirements for blot/gel reporting and figure preparation are described in detail at https://journals.plos.org/plosone/s/figures#loc-blot-and-gel-reporting-requirements and https://journals.plos.org/plosone/s/figures#loc-preparing-figures-from-image-files. 

The original gel images will be uploaded to the University of Cambridge Apollo repository and linked to the paper. They will be in that repository and a link will be provided once the manuscript is accepted. 

4. When you submit your revised manuscript, please ensure that your figures adhere fully to these guidelines and provide the original underlying images for all blot or gel data reported in your submission. See the following link for instructions on providing the original image data: https://journals.plos.org/plosone/s/figures#loc-original-images-for-blots-and-gels. 

Figures have been checked, and figure 1A and 1B combined into a single Fig1. All figures were passed through the PACE tool for resizing and resolution. 

The original gel images will be uploaded to the University of Cambridge Apollo repository and linked to the paper. They will be in that repository and a link will be provided once the manuscript is accepted. Sentence explaining this has been added to the cover letter. 

Reference list has been checked, and ensured to be complete and correct to the best of our knowledge at this time. Citation style has been altered to include square brackets instead of round brackets as per the journals guidelines.

---

## [Decision Letter · Decision Letter 1]

22 Nov 2023

Characterisation Of The R2R3 Myb Subgroup 9 Family Of Transcription Factors In Tomato

PONE-D-23-25238R1

Dear Dr. Glover,

We’re pleased to inform you that your manuscript has been judged scientifically suitable for publication and will be formally accepted for publication once it meets all outstanding technical requirements.

Kind regards,

Nguyen Hoai Nguyen

Academic Editor

PLOS ONE

Additional Editor Comments (optional):

One of the Reviewer's comments is still concerning on the Reference list format.

The authors have to complete the formatting as per the Journal's instructions.

Reviewers' comments:

Reviewer's Responses to Questions

**Comments to the Author**

1. If the authors have adequately addressed your comments raised in a previous round of review and you feel that this manuscript is now acceptable for publication, you may indicate that here to bypass the “Comments to the Author” section, enter your conflict of interest statement in the “Confidential to Editor” section, and submit your "Accept" recommendation.

Reviewer #1: All comments have been addressed

Reviewer #2: All comments have been addressed

2. Is the manuscript technically sound, and do the data support the conclusions?

Reviewer #1: Yes

Reviewer #2: Yes

3. Has the statistical analysis been performed appropriately and rigorously? 

Reviewer #1: Yes

Reviewer #2: Yes

4. Have the authors made all data underlying the findings in their manuscript fully available?

Reviewer #1: Yes

Reviewer #2: Yes

5. Is the manuscript presented in an intelligible fashion and written in standard English?

Reviewer #1: Yes

Reviewer #2: Yes

6. Review Comments to the Author

Reviewer #1: The paper provides an engaging and innovative exploration of the R2R3 Myb Subgroup 9 transcription factor family within the context of epidermal cell outgrowths in tomato. It lays a robust foundation by offering a comprehensive background that underscores the pivotal role of these transcription factors in plant development, effectively framing the study's significance. The methodology section is succinct yet informative, clearly outlining the research approach. Furthermore, the content effectively highlights the observed variation in epidermal cell outgrowth across different genes, lending depth to the investigation. The identification of SlMIXTA-2 and SlMYB17-2 as potential regulators of specific trichome types is a promising finding, poised to drive further research into their respective roles. The writing maintains a high level of clarity, precision, and scientific rigor throughout.

I have no comment anymore. I am satisfied with all the revises.

Reviewer #2: All my comments have been addressed in the revised manuscript except the format of references still do not similar with the main text. This manuscript can be accepted if the authors correct this point.

7. PLOS authors have the option to publish the peer review history of their article (what does this mean?). If published, this will include your full peer review and any attached files.

Reviewer #1: No

Reviewer #2: No

---

## [Editor Report · Acceptance letter]

14 Dec 2023

PONE-D-23-25238R1 

PLOS ONE

Dear Dr. Glover, 

I'm pleased to inform you that your manuscript has been deemed suitable for publication in PLOS ONE. Congratulations! Your manuscript is now being handed over to our production team.

Kind regards, 

on behalf of

Dr. Nguyen Hoai Nguyen 

Academic Editor

PLOS ONE